# Occupational Etiology of Oropharyngeal Cancer: A Literature Review

**DOI:** 10.3390/ijerph20217020

**Published:** 2023-11-03

**Authors:** Rayan Nikkilä, Suvi Tolonen, Tuula Salo, Timo Carpén, Eero Pukkala, Antti Mäkitie

**Affiliations:** 1Department of Otorhinolaryngology—Head and Neck Surgery, HUS Helsinki University Hospital, University of Helsinki, FI-00029 Helsinki, Finland; 2Finnish Cancer Registry, Institute for Statistical and Epidemiological Cancer and Research, FI-00139 Helsinki, Finland; 3Research Program in Systems Oncology, Faculty of Medicine, University of Helsinki, FI-00014 Helsinki, Finland; 4Department of Oral and Maxillofacial Diseases, Clinicum, University of Helsinki, FI-00014 Helsinki, Finland; 5Translational Immunology Research Program (TRIMM), University of Helsinki, FI-00014 Helsinki, Finland; 6Research Unit of Population Health, University of Oulu, FI-90014 Oulu, Finland; 7Medical Research Centre Oulu, Oulu University Hospital, University of Oulu, FI-90220 Oulu, Finland; 8Department of Pathology, HUS Helsinki University Hospital, University of Helsinki, FI-00029 Helsinki, Finland; 9Health Sciences Unit, Faculty of Social Sciences, Tampere University, FI-33014 Tampere, Finland; 10Division of Ear, Nose and Throat Diseases, Department of Clinical Sciences, Intervention and Technology, Karolinska Hospital, Karolinska Institutet, SE-17177 Stockholm, Sweden

**Keywords:** head and neck cancer, exposure, occupation, OPC, oropharynx, occupational

## Abstract

While abundant evidence exists linking alcohol, tobacco, and HPV infection to a carcinogenic impact on the oropharynx, the contribution of inhalational workplace hazards remains ill-defined. We aim to determine whether the literature reveals occupational environments at a higher-than-average risk of developing oropharyngeal cancer (OPC) and summarize the available data. To identify studies assessing the relationship between occupational exposure and risk of OPC, a search of the literature through the PubMed-NCBI database was carried out and, ultimately, 15 original articles meeting eligibility criteria were selected. Only original articles in English focusing on the association between occupational exposure and risk or death of specifically OPC were included. The available data are supportive of a potentially increased risk of OPC in waiters, cooks and stewards, artistic workers, poultry and meat workers, mechanics, and World Trade Center responders exposed to dust. However, the available literature on occupation-related OPC is limited. To identify occupational categories at risk, large cohorts with long follow-ups are needed. Identification of causal associations with occupation-related factors would require dose–response analyses adequately adjusted for confounders.

## 1. Introduction

Oropharyngeal carcinoma (OPC), comprises malignancies located in the base and posterior one-third of the tongue, the tonsils, soft palate, and posterior and lateral pharyngeal walls [1]. According to the Global Cancer Observatory [2], approximately 100,000 new cases of OPC were diagnosed worldwide in 2020. Globally, age-standardized incidence rates per 100,000 vary across regions from 0.3 to 4.4 among men and 0.1 to 0.7 among women. The wide variation in incidence rates reflects variations in the prevalence of risk factors such as tobacco smoking, alcohol consumption, areca nut chewing, and human papillomavirus (HPV) [3]. As opposed to chemical-associated tumors, HPV-driven OPCs constitute a separate entity, characterized by distinct molecular alterations, better response to therapy, and improved outcomes [4,5].

Despite declining smoking rates and alcohol use, the incidence of OPC is increasing in both old and young men in several countries [6,7]. For instance, an annual 2.7% increase among men was noted in the US between 2001 and 2017 [7]. In recent decades, research has particularly highlighted the surge in the proportion of HPV-driven OPCs [8,9]. Of the several strains of human papillomavirus, HPV16 is the most prevalent in OPC [10,11]. Occupational and environmental toxins have also been investigated for a potential role in carcinogenesis and for a hypothetical contribution to these observed increasing temporal trends [12].

While abundant evidence exists linking alcohol, tobacco smoking, and HPV infection to a carcinogenic impact on the oropharynx, the contribution of workplace hazards remains ill-defined [3]. We aim to determine whether the literature reveals occupational environments at higher-than-average risk of developing OPC and summarize the evidence available in addition to the NOCCA data [13]. This scoping review focuses on the body of existing research exploring occupational risk factors for OPC.

## 2. Materials and Methods

The current study complies with the PRISMA guidelines for scoping reviews. This review was not registered beforehand and no review protocol for this specific topic exists.

### 2.1. Study Selection

To identify studies assessing the relationship between occupational exposure and risk of OPC, a comprehensive electronic search of the literature through the PubMed-NCBI database was carried out in July and August 2023 (most recent search being on 28 August 2023). The search was conducted using medical subject headings (MeSH) and the following combinations of keywords: “oropharyngeal cancer” or “oropharynx AND cancer” in combination with the terms “occupation”, “profession”, “cancer incidence AND occupation”, “occupational disease”, or “work exposure”. No year constraints were employed. The search produced a total of 547 articles. After screening the titles and abstracts, 128 articles with relevance to the subject and published in the English language were retrieved for further detailed evaluation. Subsequently, only original quantitative studies focusing on the association between occupational exposure and risk or death of specifically OPC were eligible for inclusion. Studies had to convey results distinctively for the oropharynx. Single case reports, letters, and studies not analyzing humans were excluded. No restrictions on sample or cohort sizes were set. After the removal of duplicates and overlapping cohorts, 15 original articles that met eligibility criteria remained. The reference lists of these 15 articles selected were also screened.

### 2.2. Data Charting

Data charting was performed by the first author. Abstracted data included: first author, year of publication, location of the study, information on study design, sample details, occupational exposure or occupational group under evaluation, risk of OPC associated with the occupational exposure or setting under evaluation, information on confounding factors, and possible biases. Studies were evaluated for scientific rigor—methodology, analysis, interpretation of results—and the results were synthetized categorically.

## 3. Results

Table 1 summarizes the characteristics of the 15 studies included in this review.

### 3.1. Hospitality Industry

The study by Nikkilä et al. [14] conveyed an elevated incidence of OPC for male waiters (standardized incidence ratio (SIR) = 6.28, 95% CI: 4.68–8.26) and cooks and stewards (SIR 2.64, 95% CI: 1.83–3.69) when compared to the respective country-, age-, sex-, and calendar-period-specific incidence rates in the general population. A statistically significant excess risk of OPC was also observed in female waiters (SIR = 2.02, 95% CI: 1.41–2.81). The SIR for female cooks and stewards was 0.97 (95% CI: 0.53–1.62). The estimates were based on data from the Nordic OCcupational CAncer (NOCCA) project, a follow-up study of 15 million people between 1961 and 2005 in the Nordic countries (Denmark, Finland, Iceland, Norway, and Sweden) [13].

### 3.2. Welders

Welding fumes are formed when a metal is heated above its melting point and then vaporizes and condenses into very fine particles. Despite robust epidemiological evidence linking exposure to welding fumes to lung carcinogenesis, there is currently a paucity of data reporting on the association between welding fumes and OPC [15,16]. Drawing on data from a multicenter case–control study conducted during 2001–2007 (Investigation of occupational and environmental CAuses of REspiratory cancer, ICARE), Barul et al. [17] analyzed the association between welding and the risk of head and neck cancer (HNC) in men. The study included 449 cases of OPC and 2703 controls from the same geographical areas and frequency-matched for sex and age. All cancers had been initially identified from cancer registries. Welding was defined as having ever worked as a “welder or flame-cutter” or having a welding activity of more than 5% in at least one job. The multivariate analysis provided no evidence for an increased risk of OPC (odds ratio (OR) = 0.96, 95% CI: 0.67–1.38). The risk estimates were adjusted for age, area of residence, alcohol consumption, smoking habits and duration, and asbestos exposure.

### 3.3. Mechanical Wood Processing

Wood dust is classified as a human carcinogen by the International Agency for Research on Cancer (IARC) [18]. Namely, occupational exposure and prolonged inhalation of wood dust have been associated with an increased risk of developing nasopharyngeal cancer [19]. Sawmilling, forestry, carpentry, and manufacture of wood products are sectors linked with high wood dust exposure [20]. A case–control study conducted in Serbia [21] demonstrated an association between wood dust exposure and risk of OPC. The study consisted of 100 cases of OPC (89 men and 11 women) diagnosed consecutively during 1998–2000 and 100 controls treated during the same period for non-cancerous diseases of the head and neck (most frequently nasopharyngitis, sinusitis, rhinitis, and pharyngitis) matched for age, sex, and place of residence. The authors’ multivariate logistic regression analysis conveyed an OR for being ever exposed to wood dust of 4.16 (95% CI: 1.45–11.91). The model included smoking, alcohol consumption, other dental diseases, herpes simplex virus infection, and occupational exposure to dry air (all variables related to OPC at a significant level *p* ≤ 0.01 in univariate analysis). Aside from the small sample size and thus limited power, the degree of exposure was not measured; therefore, it is intricate to contextualize the observations. Furthermore, all controls were patients treated for non-cancerous head and neck diseases, which could also be associated with wood dust exposure [22]. Hence, this selection bias might have attenuated the observed OR [23]. The epidemiological data so far available are inconclusive to make any conclusive assessment between prolonged inhalation of wood dust and increased risk of OPC.

### 3.4. Agro-Industry

When handling and processing products, agro-industry workers are theoretically exposed not only to chemicals, notably nitrosamines, but also to various oncogenic viruses of cattle, sheep, and poultry [24,25,26]. A study followed up with 1527 men and 904 women employed at a soup manufacturing plant during 1950–2003 [27]. Throughout an average follow-up of 40 years during 1959–2006, four men, and no women, died from OPC, resulting in an SMR for men of 5.5 (95% CI: 1.5–14.1).

Cauvin et al. [28] explored occupational risk factors in cancers of the upper respiratory and digestive tract in men. In their case–control study comprising 667 male OPC cases and 147 male controls (healthy patients, or patients with cancer of another site, or another histological type not known to be strongly related to occupational risk factors), the authors discerned no significant association between OPC and exposure to flour. The OR for being ever exposed occupationally to flour adjusted for age, tobacco, alcohol, state of dentition, and other occupational exposures was 0.22 (95% CI: 0.07–0.68).

Johnson et al. [29] explored the mortality rates from cancer in the poultry industry. The study followed up with 1371 male and 1209 female workers of six plants involved exclusively in poultry slaughtering and processing during 1954–1979. In a follow-up until 2003, three deaths from OPC were registered (all in men) resulting in an SMR of 4.6 (95% CI: 1.0–13.5) when compared with the general population.

In a later study, Johnson et al. [30] examined a cohort of 6795 male and 3906 female meat cutters and meat wrappers who worked in supermarkets anytime between 1950 and 1979. Workers were subjected to follow-up for an average of 37 years. A total of 4270 deaths (40%) were recorded during 1950–2006. Significant excess mortality from OPC was seen in women (4 cases, SMR = 7.3, 95% Cl: 2.0–18.7), but not in men (4 cases, SMR = 1.7, 95% CI: 0.5–4.3). The authors postulated a viral hypothesis for the excess mortality observed.

### 3.5. Asphalt Industry

Suspicions of elevated cancer risk in workers of an Italian factory producing asphalt roofing led Zanardi et al. [31] to conduct an occupational cohort study by comparing two subsets of workers. It was recognized that workers at the factory during 1964–1979 were exposed particularly to asbestos. After the plant was dismissed due to a fire, a new factory with modern safety standards was established; asbestos was also eliminated from all production lines. The authors compared mortality rates and causes of death between workers employed any time before June 1979 (old factory, 10.6 working years on average) and workers hired after June 1979 (3.6 working years on average). A total of 26 (25%) and 3 (7%) deaths were registered in blue-collar workers of the first (*n* = 104) and second subset (*n* = 41), respectively, during follow-up between 1979 and 2001. Amidst the 104 blue-collar workers of the first subset exposed to the production line where asbestos was used, two deaths from OPC and two deaths from pharyngeal cancer (not indicated more precisely whether oropharyngeal or hypopharyngeal cancer) occurred, resulting in an SMR of 21 (95% CI: 8.8–51) for pharyngeal/tonsillar carcinoma. No cases of pharyngeal or OPC (expected 0.02) emerged among the second subset. The authors argued that the magnitude of the SMR strongly suggests that the excess occurrence of OPC may have been induced by carcinogenic exposure. However, employees hired after June 1979 had worked only an average of 3.6 years therefore reducing exposure time to potential carcinogens. Furthermore, only 7% of workers of the second cohort were deceased by the end of the follow-up.

### 3.6. Mechanics

From the countless conceivably carcinogenic hazards, exposure to benzine exhaust and asbestos-laden brakes is well documented for vehicle mechanics [32,33]. A Brazilian study recently published showed an elevated risk of OPC for mechanics (OR = 1.84, 95% CI: 1.66–2.11) when compared with the general population [34]. The large population-based series comprised 3095 mechanics and 123,556 individuals from the South and Southeast regions of Brazil who died of cancer between 2006 and 2017. A total of 274 (8.5%) and 6631 (5.1%) cases of death due to OPC were registered among the mechanics and the general population, respectively. The cohort was not restricted to vehicle mechanics but included all individuals whose occupation was officially coded as a mechanic, such as machine mechanics, irrespective of the duration of employment. Data on deaths were obtained from the National Mortality Information System, which according to the authors, possesses nearly perfect coverage and the proportion of imprecise causes of death remains relatively small (5.3% and 9.5% in the regions under analysis). Even though the estimates were not adjusted for confounders, notably tobacco smoking and alcohol consumption, the mortality ORs were stratified by age, race, education, and region. While the level of education among mechanics and the general distribution were unalike (*p* < 0.001), mechanics of all educational levels faced higher mortality rates from OPC than the general population. OPC mortality ORs for mechanics displayed also significantly elevated rates in all other strata.

### 3.7. Leather Industry

IARC classifies leather dust as carcinogenic to humans. Indeed, exposure to leather dust has been associated with nasopharyngeal cancer, especially in workers active in the boot and shoe industry. Radoï et al. [35] evaluated the risk of OPC associated with leather dust exposure in a case–control study based on the ICARE data. Occupational exposure to leather dust was assessed using job-exposure matrices accounting for probability, intensity, and frequency of exposure (the method is described in detail in a separate publication) [36]. The authors detected no elevated risk of OPC in workers exposed to leather dust (adjusted OR for being ever exposed to leather dust = 0.64, 95% CI: 0.31–1.29) when compared with controls matched for sex and age. Risk estimates were adjusted for age, sex, area of residence, socioeconomic status, tobacco, and alcohol. Furthermore, as the authors noted, participation rates were satisfactory (80.6% for controls, and 82.5% for cases) which curtailed ascertainment bias.

### 3.8. Printing Industry

Half a century ago in London, after a report hinting at a high incidence of bladder cancer at a newspaper printing factory, a proportional mortality study was carried out to further investigate these findings [37]. The death certificates of 670 workers relating to the years 1954–1966 were obtained and analyzed. Nearly all workers had been employed all their lives at the printing factory. Smoking habits were not known. OPC, more precisely tonsil carcinoma, was the cause of death in two workers. The expected number in the general population would have been 0.26 (reported *p*-value < 0.001).

### 3.9. Workers Exposed to Solvents

Using the ICARE database previously mentioned, Barul et al. analyzed the risk of HNC in men occupationally exposed to chlorinated [38], petroleum-based, and oxygenated solvents [39] in two distinctive studies, which included a total of 502 cases of OPC exposed to chlorinated solvents and 543 cases of OPC exposed to petroleum-based and oxygenated solvents. Controls (2780 for cases exposed to chlorinated solvents and 2738 for cases of petroleum-based and oxygenated solvents) were frequency-matched for sex and age and were comparable to the general population in relation to the prevalence of smoking, alcohol consumption, and socioeconomic status. Duration of exposure and cumulative exposure index were assessed by job-exposure matrices [36]. A corresponding study with the same methodology and based on the ICARE database examined the risk of HNC in women exposed occupationally to the aforementioned solvents; 111 cases of OPC were included [40]. Based on the results of the multivariate analysis (adjusted for residence area, tobacco, alcohol, and asbestos when examining chlorinated solvents), it can be concluded that occupational exposure to chlorinated, petroleum-based, or oxygenated solvents plays at most a trivial role in the development of OPC.

### 3.10. 9/11 World Trade Center Responders

Graber et al. [5] evaluated site-specific HNC incidence among 33,809 WTC responders during the first 11 years after the September 11 attacks in 2001. OPC revealed a significantly elevated SIR of 1.73 (95% CI: 1.02–2.73) for the years 2009–2012 when compared with the expected number of cases for the general population accounting for age, sex, ethnic group, and year. For the years 2003–2008, no significantly elevated rate was discerned, which could be expected given the long latency period of most neoplasms. Besides OPC and laryngeal cancer, no excess cancer was seen across the other anatomical subsites prompting the authors to postulate a synergistic effect of WTC dust and HPV. Namely, tissue damage and chronic inflammation caused by the WTC could facilitate HPV infection and chronicity.

### 3.11. Other Occupational Categories

The study by Nikkilä et al. [14] revealed a high excess risk of OPC among Nordic male artistic workers (SIR = 2.97, 95% CI: 2.31–3.76), seamen (SIR = 2.30, 95% CI: 1.91–2.77), and journalists (SIR = 2.09, 95% CI: 1.33–3.14). Among Nordic women, a statistically significant excess risk of OPC was observed in packers, loaders, and warehouse workers (SIR = 1.73, 95% CI: 1.07–2.64). For female artistic workers, the SIR was 2.13 (95% CI: 0.98–4.05).
ijerph-20-07020-t001_Table 1Table 1Characteristics of 15 studies included in the literature review.AuthorYearLocationOccupational ExposureExposure AssessmentStudy DesignSampleObservationsObserved Effect for OPCRemarksBarul et al.,2017France [38]Chlorinated solvents (5 types)Duration of exposure (ever, short, intermediate, long) and cumulative exposure index (low, medium, high) assessed by job-exposure matricesCase–control studyMen only OPC diagnosed 2001–2007 (ICARE data)502 OPC cases 2780 controls from same geographical area (general population) frequency-matched for ageEver exposure to at least one chlorinated solvent: OR 0.99 (95% CI: 0.76–1.29)Adjusted for age, residence area, tobacco, alcohol, asbestos exposurePotential non-differential misclassification biasBarul et al.,2019 France [39]Petroleum-based and oxygenated solvents (10 types)Duration of exposure (ever, short, intermediate, long) and cumulative exposure index (low, medium, high) assessed by job-exposure matricesCase–control studyMen only OPC diagnosed 2001–2007 (ICARE data)543 OPC cases 2780 controls from same geographical area (general population) frequency-matched for ageHigh exposure to diethyl ether: OR 7.78 (95% CI: 1.42–42.6); no elevated risk if medium or low exposureNo significant increased risk of OPC associated with other solventsAdjusted for age, residence area, tobacco, alcohol, socioeconomic statusPotential non-differential misclassification biasBarul et al.,2020France [17]WeldersAt least on job period as “welder and flame-cutter” or welding activity amounting to at least 5% of the working time in at least one jobCase–control studyMen only OPC diagnosed 2001–2007 (ICARE data)472 OPC cases2703 controls from same geographical area (general population) frequency-matched for ageWelding OR 0.96 (95% CI: 0.67–1.38)>10 years of welding: OR 1.04 (95% CI: 0.70–1.75)Adjusted for age, area of residence, tobacco, alcohol, asbestos exposurePotential recall bias, threshold for classification as welder low.Carton et al.,2017France [40]Chlorinated, petroleum-based and oxygenated solventsDuration of exposure (ever, short, intermediate, long) and cumulative exposure index (low, medium, high) assessed by job-exposure matricesCase–control studyWomen onlyOPC diagnosed 2001–2007 (ICARE data)111 OPC cases 775 controls from same geographical area (general population) frequency-matched for ageEver exposure to Perchloroethylene: OR 3.43 (95% CI: 1.01–11.8)No significant increased risk of OPC associated with other solvents10 solvents analyzedAdjusted for age, residence area, tobacco, alcoholPotential non-differential misclassification bias, small sample sizeCauvin et al.,1990France [28]Occupational exposureEver exposed to any of the 25 categories of exposure. Farmers excluded.Case–control studyMen only OPSCC diagnosed 1975–1984667 OPC cases147 controls: healthy patients, or patients with cancer of another site, or another histological type not known to be strongly related to occupational risk factorExposure to flour: OR 0.22 (95% CI: 0.07–0.68)Adjusted for age, tobacco, alcohol, state of dentition, and other occupational exposuresPotential recall biasFaramawi et al.,2015USA [27]Soup manufacturing plantWorkers identified from union rosters. No employment duration limit. Employed during 1950–2003 in Baltimore at same plant.Cohort study on mortalityReference: US general population2431 workers1527 men and 904 womenFollow-up 1959–200691,987 person-years of follow-up40 years of average follow-up1000 deaths (41%)4 deaths from OPC in men, none in womenSMR for men: 5.5 (95% CI: 1.5–14.1) SMR for women: 0.0 (95% CI: 0.0–19.5) No data on confoundersGraber et al.,2019USA [5]WTC respondersInvolved in rescue operations that followed 9/11Cohort study on incidence Reference: US general population (age- sex-, ethnic group- and year of specific cancer rates used)33,809 WTC responders30,139 men and 4948 womenFollow-up 2003–201232 cases of OPCSIR during 2003–2008: 0.90 (95% CI: 0.49–1.50) SIR during 2009–2012: 1.73 (95% CI: 1.02–2.73) Potential surveillance bias (participation in the study may result in earlier cancer diagnosis than in the general population), short follow-upGreenberg1972 UK [37]Printing factoryPrinting factory workers in Greater London whose death certificates were retrieved and analyzed. Nearly all worked all their life at the printing factoryProportional cohort study on mortalityMen onlyReference: Deaths among general population in greater London 670 workers who died during 1954–19662 OPC (tonsillar carcinoma) deathsProportionate mortality ratio 7.7 (95% CI 0.4–36)No data on confoundersJohnson et al.,2010USA [29]Poultry slaughtering and processing workersSubjects identified from union rosters. Worked exclusively in six poultry plants during 1954–1979Cohort study on mortalityReference: US general population2580 workers1371 men and 1209 womenFollow-up 1954–200386,407 person-years3 deaths from OPC recorded in men, none in womenSMR for all: 3.7 (95% CI: 0.8–10.8)SMR for men: 4.6 (95% CI: 1.0–13.5) No data on confoundersJohnson et al.,2015USA [30]Meat cutters and wrappers at supermarketsSubjects identified from union rosters. Worked anytime 1950–1979 in the meat and deli departments of supermarketsCohort study on mortalityReference: US general population10,701 workers6795 men and 3906 womenFollow-up 1950–2006299,295 person-yearsAverage follow-up 37.3 years4270 deaths (40%) 4 deaths from OPC recorded in men and 4 in womenSMR for all: 2.7 (95% CI: 1.2–5.3) SMR for men: 1.7 (95% CI: 0.5–4.3)SMR for women: 7.3 (95% CI: 2.0–18.7) No data on confoundersNikkilä et al.,2023Denmark, Iceland, Finland, Iceland, Norway, and Sweden [14]Occupational title53 occupational categoriesCohort study on incidenceReference: Country’s general population14.9 million peopleFollow-up 1961–20056155 OPC casesIn men:SIR for waiters: 6.28 (95% CI 4.68–8.26)SIR for artistic workers: 2.97 (95% CI 2.31–3.76)SIR for cooks and stewards: 2.64 (95% CI 1.83–3.69)SIR for seamen: 2.30 (95% CI 1.91–2.17)SIR for journalists (SIR 2.09, 95% CI: 1.33–3.14)SIR for economically inactive (SIR 1.92, 95% CI: 1.73–2.12)In women:SIR for waiters: 2.02 (95%CI 1.41–2.81)SIR for packers: 1.73 (95% CI 1.07–2.64)No data on confoundersRadoï et al.,2019France [35]Leather dustDuration of exposure (ever, short, intermediate, long) and cumulative exposure index (low, medium, high) assessed by job-exposure matricesCase–control study OPC diagnosed 2001–2007 (ICARE data)658 OPC cases 3555 controls from same geographical area (general population) frequency-matched for sex and ageEver exposed to leather dust: OR 0.64 (95% CI: 0.31–1.29)>7 years of exposure: OR 0.69 (95% CI: 0.22–2.16)Adjusted for age, sex, area of residence, socioeconomic status, tobacco, and alcoholPotential non-differential misclassification biasSantos et al.,2020Brazil [34]MechanicsIndividuals whose occupation was coded as mechanic in national databaseCohort study on mortalityMen only3095 mechanics who died from cancer 2006–2017274 cases of death from OPC recorded (8.5%)General population as comparison group: 123,556 cancer deaths and 6631 deaths from OPC (5.1%)OR for all: 1.84 (95% CI: 1.66–2.11)OR elevated in all race, education, and region groupsNo data on confoundersVlajinac et al.,2006Serbia [21]Wood dustExposure assessed by asking whether ever exposedCase–control studyOPC diagnosed 1998–2000100 cases of OPC 89 men and 11 women100 controls selected among patients treated during the same period for non-malignant diseases of the head and neck and (most frequently nasopharyngitis, sinusitis, rhinitis, and pharyngitis) matched for sex, age, and place of residenceOR 4.16 (95% CI: 1.45–11.91)Adjusted for smoking, alcohol consumption, other dental diseases, herpes simplex virus infection, occupation exposure to dry air, and smoking x alcohol consumptionSelection bias (controls patients treated for non-cancerous head and neck diseases), small sample sizeZanardi et al.,2013Italy [31]Asphalt roofing factory workers exposed to asbestosTwo subsets: (1) All workers employed at factory using asbestos 1964–1979 until factory was closed. (2) Workers employed after 1979 and not exposed to asbestos.Cohort study on mortalityMen onlyReference: General population from same region104 blue-collar workers exposed to production line employed 1964–1979 when asbestos was used (10.6 average working years)41 workers exposed to production line employed after 1979 (3.6 average working years)Follow-up 1964–20012 deaths from OPC (palatine tonsil) and 2 from pharyngeal cancer (i.e., either oropharyngeal or hypopharyngeal) recorded in production line workers exposed to asbestos (expected < 0.2)No deaths from OPC in workers employed after 1979 (expected < 0.02)SMR of lip, oral, and pharyngeal cancer for production line workers exposed to asbestos: 21.1 (95% CI: 8.8–50.7)Short exposure time among workers employed after 1979Abbreviations: 95% CI, 95% Confidence Interval; HNC, Head and Neck Cancer; ICARE, Investigation of occupational and environmental CAuses of REspiratory cancer; OPC, Oropharyngeal cancer; OPSCC, Oropharyngeal Squamous Cell Carcinoma; OR, Odds Ratio; SCC, Squamous Cell Carcinoma; SIR, Standardized Incidence Ratio; SMR, Standardized Mortality Ratio; WTC, World Trade Center.

## 4. Discussion

Our scoping review consists of only 15 original studies on occupational risks of OPC, which underlines the important knowledge gaps in this topic. Currently, the evidence is insufficient to ensure sound conclusions and the lack of robust quantitative data hinders us from presenting specific time-response relationships. Nonetheless, based on the available literature, certain hypotheses can be advanced which could point to novel directions of research. Indeed, our review hints at a plausible increased risk of OPC in certain occupational settings: in waiters, cooks and stewards, artistic workers, poultry and meat industry, soup manufacturing, mechanics, and WTC responders exposed to dust after the September 11 attacks. However, the evidence stems from a few studies, some of which are hampered by methodological flaws.

Many published analyses are inherently underpowered to detect significant associations. Particularly in studies comprising all HNCs, group analyses by cancer site were generally restricted to a small number of events. In the French studies based on ICARE data [17,35,38,39,40], occupational activity was self-reported which may result in incorrect estimates [41]. Self-report bias may also affect the studies by Cauvin et al. [28] and Vlajinac et al. [21], as exposures under evaluation were also self-reported. Furthermore, one can claim that a degree of mobility in the labor force can lead to significant changes in exposure levels. Moreover, as previously mentioned, the case–control study by Vlajinac et al. [21] may be affected by selection bias as controls were patients recruited from within the same institution treated for non-cancerous head and neck diseases, which could also be linked to wood dust exposure [23]. In the study by Graber et al. [5] follow-up of WTC was limited to less than 10 years. Given the long latency between exposure and the occurrence of cancer, any meaningful relationship may not manifest during follow-up, and thus risk estimates may be even higher.

Most importantly, inadequate adjustment for confounding factors prevents reaching valid conclusions. Most case–control studies did adjust their risk estimates for tobacco smoking and alcohol drinking. However, several potential confounders were not accounted for. A limitation present in several studies is the potential confounding from the general health status of the evaluated workers. For instance, studies have demonstrated the confounding effect of BMI on HNC outcomes [42,43,44]. BMI was considered in only one case–control study [21]. The absence of details on health status, nutrition, and other significant habits may entangle the interpretation of the results.

Failure to account for the presence of effect modification can also bias study results and lead to erroneous conclusions [45]. In certain analyses, stratification suggested the presence of potential effect modification. For instance, in the study of Johnson et al. [30] investigating cancer mortality in meat cutters and wrappers, the risk was elevated in women but not in men. The authors hypothesized that since commonly only women were employed as meat wrappers (and men conversely as meat cutters) they were exposed to carcinogens contained in the fumes from the wrapping machines, which subsequently led to a higher cancer incidence. However, for workers in the poultry industry, the risk for OPC was only elevated in the male stratum [29]. A caveat also in this regard centers on the modifying effect of socioeconomic status, which may distort results and was not considered in most studies.

It can be speculated that a mix of occupational and non-occupational exposures to various carcinogens from multiple sources, along with socioeconomic factors, may synergistically lead to a hazardous cancer burden. In parallel, little doubt exists that the complexity of confounding factors prevailing in many occupational environments may incur chance associations with OPC. Studies have demonstrated that individuals with more deprived socioeconomic status (marital status as single, education less than high school completion, and annual household income of less than USD 20,000) have a higher incidence of HNC, even after controlling for smoking and alcohol consumption [46,47]. Concerning specifically OPC, while HPV-negative OPC seems significantly more common in lower socioeconomic groups, HPV-positive oropharyngeal tumors have been associated with higher socioeconomic status [48]. Thus, it would be difficult to judge with certainty whether some individuals are more at risk owing to their occupation or their socioeconomic class. One can reasonably argue that some occupations attract workers from certain classes, hence social class can determine occupation. Lower socioeconomic status may also worsen survival in cancer, which may skew upward the results of studies examining mortality from cancer [49,50]. In the United States, OPC patients from lower socioeconomic backgrounds (annual household income of less than $30,000 US) have been associated with worse overall survival when compared with patients from higher socioeconomic strata [50].

Notwithstanding the absence of clearly defined associations between occupational exposures and OPC, there are indications suggesting an increased risk in a few professional groups, such as waiters, cooks and stewards, artistic workers, poultry, meat, and soup manufacturing workers, mechanics, construction industry workers, and WTC responders exposed to dust after the September 11 attacks [5,27,29,30,31,34]. The increased incidence of OPC in waiters may be due to exposure to environmental tobacco smoking, i.e., passive smoking, which in conjunction with their own smoking habits and alcohol consumption might explain the six-fold risk of OPC in this occupational group [14,51]. Exposure to cooking fumes may also have contributed to the excess risk of OPC observed, not only in male cooks and stewards but also in waiters and waitresses. The difference in risk estimates observed between male and female cooks and stewards may be due to different work assignments and thus different risk exposure profiles, such as in the study of meat cutters and wrappers by Johnson et al. [30], where a significantly increased risk of OPC was, conversely, observed only in women, which the authors argued could have been due to differences in working tasks between men and women. HPV infection, a well-known risk factor for OPC, might have also played a role in the increased risk of OPC among waiters, cooks, and stewards. As the average latency period between HPV infection and cancer occurrence has been estimated to be between 10 to 30 years, all HPV-induced OPCs may still not be observable in studies with short follow-up and, consequently, the risk estimates for certain occupational categories could even be higher with longer follow-up [52].

For the elevated risk of OPC observed in WTC responders 7 to 10 years after the attacks, Graber et al. [5] suggested a synergistic effect of dust exposure with HPV, as other HNC subsites did not show elevated rates. It would not be unreasonable to hypothesize that the excess OPC observed in certain occupations may be attributable to oncogenic viruses. Workers in poultry and meat slaughtering and processing plants who handle these animals may be continuously exposed to these viruses [29,30]. However, the dearth of epidemiologic and molecular studies confirming the presence of viruses in human tumors prevents us from assigning with certainty an etiological role to these viruses. Some evidence of a viral etiology stems from the molecular study of Ursu et al. [53]. The authors analyzed 26 fresh HNC tumor specimens (24 HNSCCs) and detected a high prevalence of known oncogenic viruses other than HPVs in 23 (88%) of the samples. Among the detected viruses were herpes simplex 6, 7, and 8, molluscum contagiosum virus, human polyomavirus 6, Epstein–Barr virus type 1 and type 2, and cytomegalovirus. Still, the presence of the viral DNA itself in the tumor tissue is not sufficient evidence of association, and hardly of causation. The existence of viruses in cancerous lesions may be incidental and solely colonization of a pre-existing cancerous lesion owing to suitable environmental conditions.

Studies have also suggested that healthcare workers treating HPV-associated conditions may experience occupational exposure to the virus. For instance, infectious oncogenic viruses can be dispersed via ablation smoke generated during surgical procedures [54]. A Finnish study evaluated the transmission risk from patients to personnel during surgical treatment of laryngeal papillomas and laser treatment of urethral warts. The authors of the study concluded that even though HPV may contaminate protective equipment, particularly surgical gloves, transmission of HPV to medical personnel during procedures is unlikely when aseptic practices are adhered to [55]. Still, anecdotal reports of HPV transmission from patients to healthcare professionals do exist [56,57].

Certain shortcomings in our review need to be mentioned. As we only retrieved publications in English from the PubMed-NCBI database, we may have omitted some studies written in other languages or located in other databases. Secondly, we did not carry out a systematic quality appraisal of the articles included.

## 5. Conclusions and Future Directions

The available published literature on occupational OPC is scarce, and the case reports of OPCs induced by occupational exposure are few and mostly anecdotal. Several studies define OPC as malignancies of the oral cavity and pharynx and therefore fail to analyze these cancers as distinct entities (the studies were not included in our review). Moreover, as safety standards may have evolved and consequently carcinogen exposures changed (for instance asbestos exposure [31]), observations described for certain industries in the literature may not reflect today’s circumstances in these same industries. Our review reiterates the need for larger cohorts with long follow-ups to identify occupations at risk of OPC. A risk of OPC appears to be elevated among restaurant waiters, cooks and stewards, artistic workers, poultry and meat workers, mechanics, and World Trade Center responders exposed to dust. Identification of causal associations with occupation-related factors would require dose–response analyses adequately adjusted for confounders.

## Data Availability

This review paper is based on 15 original articles, the details of which can be found in the reference list. The data used in this review are publicly available in these cited sources.

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
