# Peer review of "Occupational Etiology of Oropharyngeal Cancer: A Literature Review"

_ijerph, 2023, doi:10.3390/ijerph20217020_

Round 1
Reviewer 1 Report
Comments and Suggestions for Authors
1. The study is about showing increased risk for oropharyngeal cancers (OPC) among various occupational types in men and women.
2. I agree that published works showing potential risks for OPC are scarce and hence inconclusive to generate valid conclusions.
3. I suggest that in the results section where the 15 published works are tabulated, another column can be added to specify its drawbacks (such as presence of biases, other methodological flaws, inability to control for confounders etc)
4. I agree with the recommendations on the use of larger cohorts with longer follow-up to better quantify the risk for OPC.
Author Response
We are thankful to the Reviewer for the time taken to evaluate our review. Additionally, we are grateful for the suggestion. We have edited Table 1 accordingly. Due to size, we opted not to add an additional column, and instead added the information concerning biases and limitations in the fourth column under "Remarks".
Reviewer 2 Report
Comments and Suggestions for Authors
Dear authors,
Your work brings no novelty into the field of oncology in regards to the ethology of this type of cancer. There are already informations about this topic in ESMO and ASCO guidelines.
Although, for physicians who do not have access to this type of info, this review is easy to read and understand.
Thank you for understanding.
Comments on the Quality of English Languageno comments. well written
Author Response
We sincerely appreciate the time and effort invested by the Reviewer in evaluating our work. We acknowledge that our review does not necessarily present any groundbreaking novelty, but still summarizes the available data.
Reviewer 3 Report
Comments and Suggestions for Authors
This is an interesting and important work, Authors have covered a significant number of publications and done a hard work to identify the relationship of occupational exposure and oro-phrayngial carcinoma. Strength of the article will be greater if it has followed the standard method of a scoping review or a systematic review. Details that they have included and the efforts are very closer to a scoping review. It will not qualify as a narrative review. Authors are encouraged to re-write the same as a scoping review by modifying the methodology which will enhance the scientific validity.
Author Response
We are thankful to the Reviewer for the time taken to evaluate our review. We are truly grateful for the suggestion. We have edited the manuscript (mostly the Methods section) according to PRISMA guidelines for scoping reviews.
Round 2
Reviewer 3 Report
Comments and Suggestions for Authors
Authors have not done the requested changes